# A Large Piezoelectric Strain Recorded in BCT Ceramics Obtained by a Modified Pechini Method

**DOI:** 10.3390/ma13071620

**Published:** 2020-04-01

**Authors:** Lucjan Kozielski, Agnieszka Wilk, Mirosław M. Bućko, Juras Banys

**Affiliations:** 1Faculty of Science and Technology, University of Silesia, 1A 75 Pułku Piechoty St., 41-500 Chorzów, Poland; 2AGH–University of Science and Technology, Faculty of Materials Science and Ceramics, al. Mickiewicza 30, 30-059 Krakow, Poland; Agnieszka.Wilk@agh.edu.pl (A.W.); bucko@agh.edu.pl (M.M.B.); 3Faculty of Physics, Vilnius University, Saulėtekio 9/3, LT-10222 Vilnius, Lithuania; juras.banys@ff.vu.lt

**Keywords:** piezoelectric materials, ferroelectrics, lead-free ceramics

## Abstract

There is a strong need in the industry to develop lead-free piezoelectrics for sensors and actuators. Although these materials have become an important component of many electronic devices, it is very important for the industry to decarbonise ceramic technology, especially through the introduction of modern sintering technologies. Among the many piezoelectric compounds available, Calcium Barium Titanate (BCT) have been widely investigated because of its similar performance to lead-containing Lead Titanate Zirconate (PZT). In this paper, a modified Pechini method for obtaining ceramic Ba_0.9_Ca_0.1_TiO_3_ nano-powders is described. Deviation from the established procedure resulted in the precipitation of the solution or obtaining of a low-quality (poorly crystallized) product with numerous impurities. The samples of BCT materials were examined to find their ideal microstructures and structures; these factors were confirmed by their outstanding X-ray diffraction spectra and high piezoelectric constant values that are comparable to commercial lead-containing materials.

## 1. Introduction

Piezoelectric materials are widely used in piezoelectric motors, transformers, linear actuators, and diesel engine fuel injectors [1]. The electromechanical properties of BaTiO_3_–CaTiO_3_ (BCT) composites have been widely studied because BCT materials show good composition with morphotropic phase boundary (MPB), similar to that of Lead Titanate Zirconate (PZT) [2]. Thus, BCT is a good material to replace PZTs in many piezoelectric applications [2].

Much less is known, however, about the critical role of processing on the properties of lead-free ceramics [3]. Recently, J.P. Praveen et al. showed that, for a series of lead-free ceramics, the implementation of wet chemistry technologies results in a strong decrease in dielectric loss above Tc and a dramatic enhancement in electromechanical performance [4]. The influence of oxygen vacancies on these properties was discussed; however, the ferroelectric and piezoelectric properties of these materials were not studied in detail.

The standard Mixed Oxide Method (MOM) used for the fabrication of barium titanate types of ceramics requires a high temperature for solid-state sintering and results in an inhomogeneous structure, which strongly affects its piezoelectric performance. In contrast, the chemical method for ceramic nano-powder synthesis has the advantages of perfect chemical stoichiometry and ideal homogenous grain size distribution.

Generally, the lattice constants and grain size, which are directly dependent on the obtained ceramic nano-powder’s particle sizes and sintering conditions, have been shown to be very important in terms of optimizing the piezoelectric properties of BCT materials [5,6,7]. There are very few scientific reports on the synthesis of BCT ceramics using the Pechini method [8,9,10]. 

The Pechini method is a variation of the sol-gel technique dedicated to simple titanates and niobates of lead and alkali metals, mainly using nitrates and alcoholates as reagents [11].

It should be emphasized, however, that for more complex compounds with a greater number of cations, the course of the whole process is complicated, and the effects of the carboxylic acid and polyhydroxyl alcohol used, as well as the molar ratio between them, are not fully understood. The nature of the formed gel and polymer resin, the stability of the chelates, and the type of connections between them are also unknown. There is also no information on the effects of calcination conditions and the specific effects of the obtained precursors on the final reaction products [12]. For this reason, the Pechini technique has been modified by experimental trial and error to obtain a more complex composition. In this work, the analyzed compound is made up of three cations of different chemical natures (Ti—acid, Ba, and Ca—alkaline). As a result, these cations condensed, hydrolyzed, and polymerized completely differently. For this reason, it is difficult to obtain a stable solution that is a homogenous mixture of chelated Ba, Ca, and Ti cations. Moreover, as the experiments using nitrates did not produce the expected results, the authors decided to use carbonates as the sources of the Ba and Ca cations.

The final dielectric, piezoelectric, and ferroelectric properties of the synthesized BCT were measured and correlated with the structure and preparation method in this work.

The calculated results enabled the authors to reveal the unexpectedly high dielectric and ferroelectric properties of the synthesized BCT, as detected by a high-resolution Piezoelectric Evaluation System. Surprisingly, the anomalous electromechanical properties still lack clarification, despite the considerable number of studies devoted to this material. Thus, the quality of the BT-based compounds used in commercially available piezoelectric transducers depends critically on their materials and processing methods.

## 2. Materials and Methods 

For the synthesis of the Ba_0.9_Ca_0.1_TiO_3_ ceramics, analytical grade chemicals were used from Sigma-Aldrich, Poznań, Poland. In this experiment, the authors determined the appropriate order and method for introducing the selected substrates into the system to obtain and maintain a stable starting solution with homogenization at the atomic level. In addition, the correct ratio of citric acid to ethylene glycol was set to eliminate carbonate contamination from the product. Any deviation from the established procedure resulted in precipitation of the solution or obtaining of a low quality (poorly crystallized) product with numerous impurities.

The starting materials used for the modified Pechini method synthesis of BCT were barium and calcium carbonate powders (BaCO_3_, CaCO_3_) and a solution of titanium (IV) isopropoxide Ti[OCH(CH_3_)_2_]_4_. The applied technology chart is presented in Figure 1.

The resultant polymer was dried for 24 h, pre-calcinated at 350 °C (4 h), and then ground to fine powder in a mortar. Calcination conditions were selected to obtain a single-phase product without impurities and unreacted substrates. The calcination temperature was as low as 700 °C for 4 h to obtain the BCT phase formation confirmed by X-ray diffraction studies. The calcined powder was compacted into pellets of 10 mm diameter and 2 mm thickness using a uniaxial press at 100 MPa. 

The final sintering step was conducted at 1400 °C for 2 h at a heating rate of 5 °C/min. The density of the sintered pellets was measured using the Archimedes’ method. A PANalyticalX’Pert Pro (Malvern Panalytical, Almelo, the Netherlands) multifunctional diffractometer was used to analyze the obtained samples at angles between 20° and 80° (Cu Ka radiation). The microstructural examination of the sample surface was done using a NOVA NANO 200 SEM (FEI Company, Hillsboro, OR, USA). Ferroelectric and piezoelectric measurements were done using a Piezoelectric Evaluation System (aixACCT GmbH, Aachen, Germany). Polarization and displacement vs. electric field measurements were carried at 10 Hz with the maximum applied electric field of ±4 kV/cm. 

## 3. Results and Discussion

The crystallite dimensions and crystalline structure parameters of BTC prepared by the modified Pechini method were determined by the X-ray diffraction analysis (XRD). As mentioned above, the calcination temperature for the Ba_0.9_Ca_0.1_TiO_3_ ceramic nano-powder obtained by the modified Pechini method was as low as 700 °C for 4 h. Figure 2a shows the X-ray diffraction spectra of the calcined BCT powder. The X-ray diffraction pattern of the powder calcinated at 700 °C has no additional peaks from the unreacted oxides (Figure 2a). This indicates complete phase formation at a low temperature. Consequently, the calcination temperature of BCT was evaluated to be as low as 700 °C for 4 h.

The calcination temperature evaluated in our experiments is ~300–500 °C lower than that of the conventional solid-state method [13]. Consequently, we developed a much more energy-efficient sintering process, based on a techno–economic analysis of the implementation of the Pechini method. This is very important in the ceramic industry, which is energy intensive; decarbonising ceramic manufacturing has necessitated more forms of efficient manufacturing, particularly through the introduction of modern sintering technologies [14].

The most important fact that emphasizes the importance of our discovery is that in the article by J.P. Praveen et al., which describes the preparation of BCT ceramics by the sol-gel chemical method, the calcining temperature used was also at a high level of 900–1100 Celsius [4].

This strongly lowered calcination temperature is possibly due to the high reactivity rate of the much smaller nano-powders obtained from the successfully modified Pechini method. The average crystalline size (nm) was calculated from the Debye Scherrer relation, as was the line broadening at half the maximum intensity of Bragg’s angle. This method allowed us to obtain a crystallite size as small as 17.5 nm, whereas in J.P. Praveen et al.’s experiment, the crystallite size was only 30 nm. 

For the synthesis of nanoparticles, the hydrothermal method is increasingly being favored globally by the industry, but the recorded average crystallite sizes of the BCT powders were also no lower than 30 nm [15].

The structure of our BCT ceramics sintered at 1400 °C for 4 h was determined by X-ray diffraction (Figure 2b). A perfect perovskite structure is evident in the XRD diffraction patterns. A tetragonal P4 mm BCT structure with lattice constants of *a* = *b* = 4.0009 Å and *c* = 4.0088 Å was detected.

The morphology studies revealed that the BCT nanopowders obtained by the modified Pechini method possessed submicron sizes and was semi-agglomerated (Figure 3a).

The SEM of the polished surface of the Ba_0.9_Ca_0.1_TiO_3_ ceramics sample revealed a perfect grain arrangement in the ceramics obtained by the investigated method (Figure 3b). The SEM micrograph of the BCT pellet sintered at 1400 °C for 4 h shows a full dense and pore-free microstructure with a homogenous grain distribution throughout the microstructure. The average grain size was estimated to be ~1–2 μm. 

Electric field dependent polarization (E-P) and displacement (ED) hysteresis loops were measured when a triangular wave shape voltage with a frequency of 1 Hz was applied to a BCT sample (Figure 4). These characteristics were collected over a temperature range from −70 to 100 centigrade. The dielectric, ferroelectric, and piezoelectric parameter values calculated from these measurements at room temperature (RT) and at T= −50 °C are listed in Table 1. The temperature of T= −50 °C was chosen because the ferroelectric properties in that region were better than those in any other region.

The BCT sample showed a remnant polarization (Pr) as high as 9.7 μC/cm^2^ at a temperature of −50 °C, parallel with a small coercive field (Ec) of 0.431 kV/cm (Figure 4a). The remnant polarization was higher for any others reported in previous papers for BCT [16]. The same parameters recorded at room temperature (RT) are Pr = 4.1 μC/cm^2^ and an Ec of 0.280 kV/cm, respectively, which also reveal an unusually high value for this particular composition.

Such a high Pr value is usually attributed to the compositional homogeneity and uniformity of grain sizes, as well as a lack of defects that act as domain-wall pinning centers [17]. 

The ferroelectric and piezoelectric measurements were recorded using a Piezoelectric Evaluation System, in order to verify the predictions obtained by the structural analysis. 

Figure 4 indicates a highly isotropic deformation of the BCT sample for subsequent field strength values (see Figure 4b), exactly as predicted by numerical simulation [18]. Interestingly, the cascade of butterfly curves that appears in the positive direction does not appear in the negative areas, which coincides with the simulated results for the isotropic polycrystalline BCT cited above. A similar conclusion emerges from Figure 4a (ferroelectric hysteresis loops).

Figure 4b presents the piezoelectric strain for the BCT samples. As large a strain as d = 0.14% was recorded under an electrical field as low as 4 kV/cm at a temperature of −50 °C. Additionally, a very small hysteresis is shown in Figure 4b for a large strain that can achieve an almost perfect structure with low ferroelectric domain pinning. The same parameter value recorded at room temperature (RT) is d = 0.14%, which is also extremely high for this particular composition [19].

The BCT sample showed a remnant polarization value Pr as high as 11.55 μC/cm^2^ at a temperature of −50 °C (Figure 5b) in parallel with the small coercive field (Ec) of 0.431 kV/cm, which is lower than the sizes reported in the literature for this particular composition [20]. The same parameters recorded at room temperature (RT) are Pr = 4 μC/cm^2^ (Figure 5b) and an Ec of 0.280 kV/cm, which are also unique values for this particular composition [21].

The piezoelectric strain coefficients (d_33_) at the chosen temperatures were calculated by measuring the displacement of the prepolarized BCT samples under the bipolar triangle driving the voltage (Table 1). Surprisingly, no significant drop of d_33_ in the composites was observed between −50 °C and RT (Figure 5a). This can be explained by the previously discussed model of isotropic polycrystalline BCT, which lacks defects that do not act as pining centers for the domain walls, resulting in a high thermal stability of the piezoelectric properties. However, the most important result of this research was the development of material with a three-fold higher piezoelectric coefficient value than the material recently reported by Salcedo-Abrair et al., who also used the latest material [22]. These piezoelectric properties are comparable to commercial lead-containing materials [23].

There was no visible difference in the level of the static dielectric permeability value in the tested BCT material compared to the values reported for similar compositions. The measured values of dielectric permeability range from 14,234 to 7618 between −50 °C and RT (Table 1). Further, as mentioned above, Wei Li et al. also recorded values from 4500 to 16,000 at various temperatures for Ba_0.1_Ca_0.9_TiO_3_ ceramics with a small addition of tin as a dielectric constant. 

## 4. Conclusions

Ceramic Ba_0.9_Ca_0.1_TiO_3_ nano-powders were obtained for the first time using the modified Pechini method. The outstanding properties possessed by the final BCT ceramics were achieved through their exceptional composition and microstructure, which required very careful control throughout the successive stages of the applied modified process.

Piezoelectric lead-free ceramics have now become an important component of many electronic devices, which is why it is very important for the industry to decarbonise ceramic technology, especially through the introduction of modern sintering technologies. The method used in this study reduced the calcination temperature by 300 degrees, compared to the methods used previously. 

Record-small 17 nm crystallites and a nearly perfect material structure were also achieved. Consequently, the final piezoelectric coefficient values were also three times higher than those recently published by Salcedo-Abrair et al. Such a high piezoelectric constant is comparable to commercial lead-containing materials.

## Figures and Tables

**Figure 1 materials-13-01620-f001:**
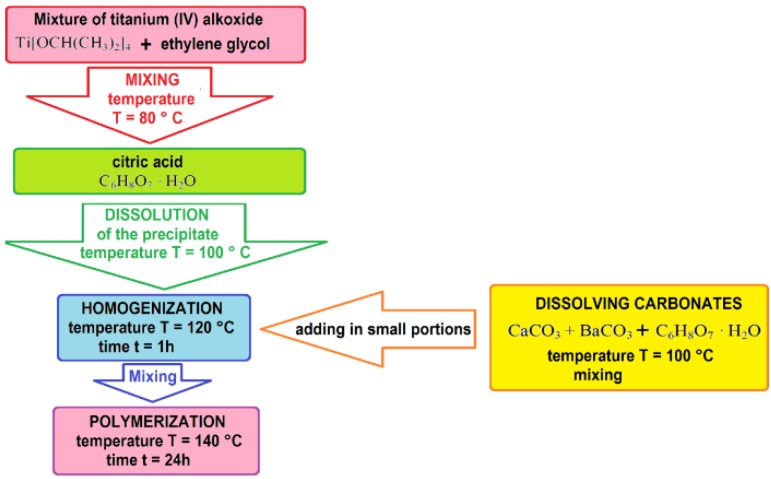
Scheme for obtaining Ba_0.9_Ca_0.1_TiO_3_ nano-powders by the Pechini method.

**Figure 2 materials-13-01620-f002:**
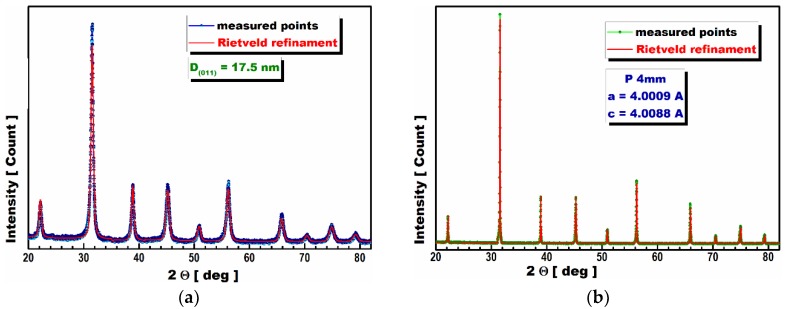
(**a**) The X-ray diffraction patterns of the calcined Ba_0.9_Ca_0.1_TiO_3_ powders; (**b**) the X-ray diffraction pattern from the Ba_0.9_Ca_0.1_TiO_3_ ceramics sample.

**Figure 3 materials-13-01620-f003:**
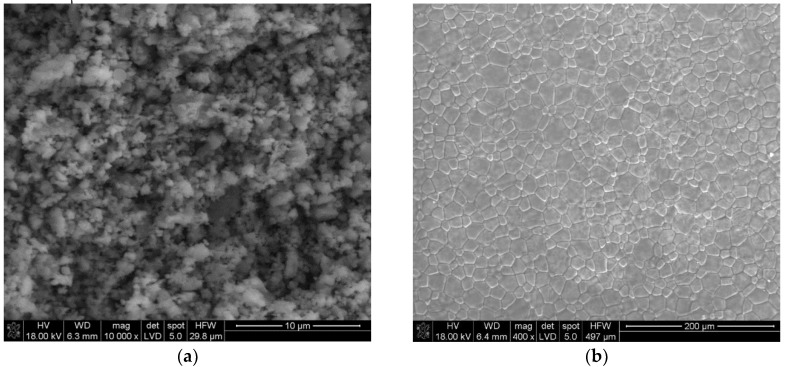
(**a**) SEM images of the nano BCT powder obtained by the modified Pechini method; (**b**) the SEM of the polished Ba_0.9_Ca_0.1_TiO_3_ ceramics’ sample surface sintered at 1400 °C for 4 h.

**Figure 4 materials-13-01620-f004:**
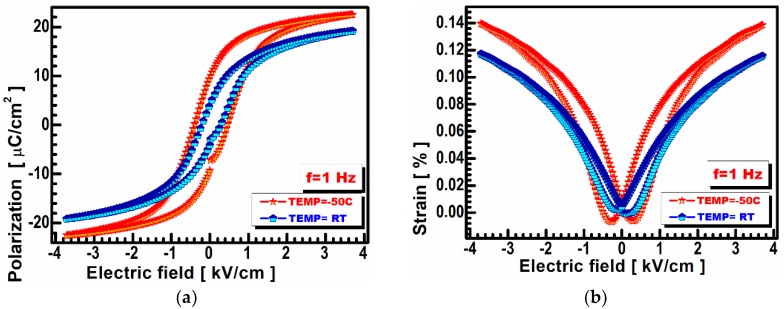
(**a**) Polarization vs. the electric field hysteresis loop at a temperature range from −70 to 100 °C; (**b**) strain vs. electric field measurements for BCT at a temperature range from −70 to 100 °C.

**Figure 5 materials-13-01620-f005:**
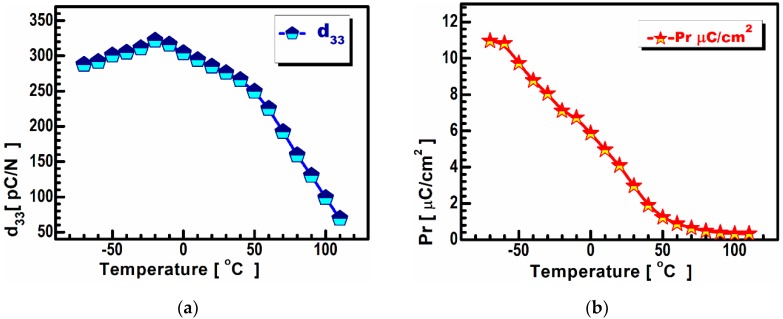
Piezoelectric coefficient vs. temperature measurement for the BCT at a temperature range from −70 to 100 °C (**a**); (**b**) polarization vs. temperature at a temperature range from −70 to 100 °C.

**Table 1 materials-13-01620-t001:** The ferroelectric and piezoelectric parameters for the obtained Ba_0.9_Ca_0.1_TiO_3_ samples.

Parameter	Unit	Value
Dielectric constant εr (at RT and f = 1 Hz)	–	7618
Dielectric constant εr (at T = −50 °C and f = 1 Hz)	–	14,234
Piezoelectric coefficient d_33_ (at RT and f = 1 Hz)	(pC/N)	285
Piezoelectric coefficient d_33_ (at T = −50 °C and f = 1 Hz)	(pC/N)	305
Remanent polarization P_r_ (at RT and f = 1 Hz)	(μC/cm^2^)	4.1
Remanent polarization P_r_ (at T = −50 °C and f = 1 Hz)	(μC/cm^2^)	9.7
Displacement (at RT and f = 1 Hz)	(%)	0.12
Displacement (at T = −50 °C and f = 1 Hz)	(%)	0.14

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
