# Peer review of "A Large Piezoelectric Strain Recorded in BCT Ceramics Obtained by a Modified Pechini Method"

_materials, 2020, doi:10.3390/ma13071620_

Round 1

Reviewer 1 Report

Revision of “Large piezoelectric strain recorded in BCT ceramics obtained by modified Pechini method”

The manuscript under review devoted to synthesis and characteristics of lead-free Ba0.9Ca0.1TiO3 (BCT) material obtained by modified Pechini method. Providing of such investigations is very important for the further application of such materials in piezoelectric motors, transformers, linear actuators or diesel engine fuel injectors etc.

In the manuscript, the author’s presents sufficiently complete information about methods for obtaining samples. It was shown that ceramic of the Ba0.9Ca0.1TiO3 have no impurity peaks and a tetragonal P4mm BCT structure with lattice constants of ? = ? = 4.0009 and ? = 4.0088 angstroms. The authors provide fairly complete data on the study of the piezoelectric properties of BCT ceramics in the temperature range from -70 to 100 centigrades, showing the dynamics of their change, which is important for practical application.

In manuscript all necessary information is captured by 4 figures and 1 table. There are 19 references, all of them are adequate and are reflected in the text.

All studies are conducted at a high scientific level, however the current manuscript not without imperfection:

  1. The authors (page 4 line 129) do not explain why the values for the table were chosen only at -50 Celsius and room temperature.
  2. In fig. 4 It is difficult to determine the temperature for each line.
  3. Since the material is interesting for practical use, it would be interesting to see a graph of the dependence of the piezoelectric properties on temperature. This is due to the fact that when designing equipment, knowledge of possible nonlinearities is required.
  4. The authors use different dimensions to indicate the electric field strength, which makes it more difficult to perceive information.
  5. Page 5 line 152 “BCT sample showed a dielectric constant …” a typo was made, because the dielectric constant is a dimensionless quantity, and polarization was meant.

The obtained results are important both for understanding the physical processes that occur in real objects and for the development of new materials. The described manuscript is sufficient, comprehensive and it corresponds to the field of the Journal «Materials». It can be accepted after revision.

Author Response

Author's Reply to the Review Report (Reviewer 1)

  1. The authors (page 4 line 129) do not explain why the values for the table were chosen only at -50 Celsius and room temperature.

The temperature T = -50 °C was chosen because the ferroelectric properties in this region were better than in any other region and additional figures 5a and 5 b were added now to present the whole range of measured temperatures:

Line 129:

The temperature T = -50 °C was chosen because the ferroelectric properties in this region were better than in any other region.

  1. In fig. 4 It is difficult to determine the temperature for each line.

To make Figure 4 clearer, the number of features presented was limited to two extreme, and additional Figure 5a displays strain connected piezoelectric coefficient vs temperature dependence and next additional Figure 5b shows the coercive field temperature characteristics

  1. Since the material is interesting for practical use, it would be interesting to see a graph of the dependence of the piezoelectric properties on temperature. This is due to the fact that when designing equipment, knowledge of possible nonlinearities is required.

The suggested missing characteristics have been completed: Additional Figure 5a displays strain connected- piezoelectric coefficient vs temperature dependence and next additional Figure 5b shows the coercive field temperature characteristics

  1. The authors use different dimensions to indicate the electric field strength, which makes it more difficult to perceive information.

The system of units used has been standardized

  1. Page 5 line 154 “BCT sample showed a dielectric constant …” a typo was made, because the dielectric constant is a dimensionless quantity, and polarization was meant.

The error was corrected in the text (yellow marked):

BCT sample showed a remnant polarization value Pr as high as high as 11.55 mC / cm2 at the temperature of – 50 °C parallel with small coercive field (Ec) of 431 V/mm.

Reviewer 2 Report

The motivation for this work and the reported results and findings do not have enough merit to qualify for scientific journal publication.
While the experimental methods are not novel, the results obtained are not that exceptional as the authors claim. There are already many published
literature on BCT ceramics with much better and interesting findings.
There are several mistakes in the text in terms of grammar, syntax, style, etc. Even the information and findings provided in the manuscript
are not sufficient enough to make it worthy to publish.

Author Response

Author's Reply to the Review Report (Reviewer 2)

The motivation for this work and the reported results and findings do not have enough merit to qualify for scientific journal publication.

 While the experimental methods are not novel,

The method of obtaining ceramic nanopowders is not really new, but it is very difficult to use for multi-component ceramic materials because a lot of factors affect the correct final composition [[1]]. Probably for this reason, it was only used 3 times for BCZT materials, but never for pure BCT material, as it happened in our case [[2],[3],[4]].

I will quote here a detailed description of the entire procedure to show the enormous effort put into properly carrying out the entire process, which took us 3 years to complete!

The Pechini method is a variation of the sol-gel technique dedicated to simple titanates and niobates of lead and alkali metals using mainly nitrates and alcoholates as reagents .

It should be emphasized, however, that in the case of more complex compounds with a greater number of cations, the course of the whole process is complicated, and the effect of the carboxylic acid and polyhydroxyl alcohol used and the molar ratio between them are not fully understood. The nature of the formed gel and polymer resin, the stability of the chelates and the type of connections in them are also unknown. There is also no information on the effect of calcination conditions and the specificity of obtained precursors  on the final reaction products . For this reason, the Pechini technique is modified by experimental trials and errors to obtain a more complex composition. In this work, the analyzed compound is made up of three cations of different chemical nature (Ti- acid, Ba, Ca-alcaline). As a result, these cations condense, hydrolyze and polymerize completely differently. For this reason, it is difficult to obtain a stable solution that is a homogenous mixture of chelated Ba, Ca and Ti cations. Moreover, as the experiments using nitrates did not bring the expected results, the authors decided to use carbonates as sources of Ba and Ca cations.

 In this experiment, the authors have determined the appropriate order and method of introducing selected substrates into the system so as to obtain and maintain a stable starting solution with homogenization at the atomic level. In addition, the correct ratio of citric acid to ethylene glycol was set to eliminate carbonate contamination from the product. Calcination conditions were selected to obtain a single-phase product without impurities and unreacted substrates. Any deviation from the established procedure resulted in precipitation of the solution or obtaining a product of low quality (poorly crystallized) with numerous impurities.

 the results obtained are not that exceptional as the authors claim.

According to the latest summary of progress in the field of lead-free materials, the largest strain in materials based on barium titanate reach only 0.2%. However, the largest elongations among published results are for very complex material structures of the KNN-modified BNT-BT system that exhibited 0.45% of unipolar strain at 80 kV/cm, but our registered elongation of 0.4% for a 3 kV / cm field is far higher than single-phase BT materials and comparable with record parameters for lead-free materials in a summary from 2019 [[5]]

There are already many published literature on BCT ceramics with much better and interesting findings.

I strongly agree with the reviewer that there are many published literature on BCT ceramics with a lot of interesting findings, but only a few focus on the lowering the synthesis temperature to offer to the industry much more economical technologies for decarbonizing our planet [[6]]. This suggests that although the ceramics sector consumes considerable energy and contributes to high emissions and environmental pollution, it remains key to delivering the global efforts towards a low-carbon economy, whilst contributing to its growth and balance. This is evident, given that much of the economic growth experienced by emerging markets today is triggered by developments in industrial and manufacturing activities that require greater resource inputs, leading to overall increase in the environmental impact of the sector. Finally, although several energy efficient sintering processes have been developed, there is no published such a techno-economic methods for barium calcium titanate ceramics.

[[1]] A. K. Rai, K.N. Rao, L. V. Kumar, K.D. Mandal: Synthesis and characterization of ultra fine barium calcium titanate, barium strontium titanate and Ba1−2xCaxSrxTiO3 (x = 0.05, 0.10), Journal of Alloys and Compounds 475 (2009) 316–320

[[2]] Y. Tian,Y Gong, D. Meng, S. Cao: Structure and electrical properties of Ir4þ-doped 0.5Ba0.9Ca0.1 TiO3–0.5BaTi0.88Zr0.12O3–0.12%La ceramics via a modified Pechini method, Materials Letters 153 (2015) 44–46

[[3]] Y. Tian, Y. Gong, D. Meng, Y. Li, B. Kuang: Dielectric Dispersion, Diffuse Phase Transition, and Electrical Properties of BCT–BZT Ceramics Sintered at a Low-Temperature, Journal of ELECTRONIC MATERIALS, Vol. 44, No. 8, 2015

[[4]] R.S. da Silva, M. I. B. Bernardi, A.C. Hernandes: Synthesis of non-agglomerated Ba0.77Ca0.23TiO3 nanopowders  by a modified polymeric precursor method, J Sol-Gel Sci Techn (2007) 42:173–179

[[5]] J. Hao, W. Li, J. Zhai, H. Chen: Progress in high-strain perovskite piezoelectric ceramics, Materials Science & Engineering R 135 (2019) 1–57

[[6]] Mahmoud A.E., Ezzeldien M., Parashar S.K.S., Enhancement of switching/un-switching leakage current and ferroelectric properties appraised by PUND method of (Ba1-xCax)TiO3 lead free piezoelectric near MPB, Solid State Sciences 93 (2019) 44-54

Reviewer 3 Report

In this manuscript, the authors reported a wet chemical method to synthesize the BCT piezoelectric ceramic powders. The folowing comments help the authors to enhance the manucsript: 

1. The writing of whole paper should be carefully checked. Many language and grammar problems can be found. 

2. The contribution and innovation of the reported technique should be cleraly stated in the manuscript. Although the authors argued that few scientific reports are about wet chemical synthesis of BCT ceramic, I could not see the merits of the reported technique from the manuscript. 

3. In addition, a comprehensive literature survey about the existing tecniques and comparions with them should be included in the introduction section. 

4. Since the key of this study is to report a BCT synthesis tecnique, I would suggest the authors to provide even more details in Materials and Methods section. 

5. Key results and conclusions should be well summarized in Conclusion section. 

Author Response

Author's Reply to the Review Report (Reviewer 3)

  1. The writing of whole paper should be carefully checked. Many language and grammar problems can be found. 

All the errors were corrected in the text (yellow marked) by “Materials” language editorial service

  1. The contribution and innovation of the reported technique should be clearly stated in the manuscript. Although the authors argued that few scientific reports are about wet chemical synthesis of BCT ceramic, I could not see the merits of the reported technique from the manuscript. 

The missing contribution and innovation of the reported technique were more clearly stated in the last part pf Introduction (yellow marked line 56 -76)

  1. In addition, a comprehensive literature survey about the existing techniques and comparisons with them should be included in the introduction section. 

All the missing literature references were fulfilled in the text (yellow marked in line 52 and 54), Generally there are very few scientific reports on the synthesis of BCT type of ceramics using wet chemical synthesis techniques

  1. Since the key of this study is to report a BCT synthesis technique, I would suggest the authors to provide even more details in Materials and Methods section

The more details in Materials and Methods section were provided (yellow marked from line 78 to 103)

  1. Key results and conclusions should be well summarized in Conclusion section

The missing key results and conclusions were summarized in details in Conclusion section

Round 2

Reviewer 3 Report

The authors have addressed all the comments properly.